# Patients with Myalgic Encephalomyelitis/Chronic Fatigue Syndrome (ME/CFS) Greatly Improved Fatigue Symptoms When Treated with Oxygen-Ozone Autohemotherapy

**DOI:** 10.3390/jcm11010029

**Published:** 2021-12-22

**Authors:** Umberto Tirelli, Marianno Franzini, Luigi Valdenassi, Sergio Pandolfi, Massimiliano Berretta, Giovanni Ricevuti, Salvatore Chirumbolo

**Affiliations:** 1Tirelli Medical Group, 33170 Pordenone, Italy; utirelli@cro.it; 2Italian Society of Oxygen Ozone Therapy (SIOOT), University of Pavia, 27100 Pavia, Italy; marianno.franzini@gmail.com (M.F.); luigi.valdenassi@unipv.it (L.V.); sergiopandolfis2@gmail.com (S.P.); giovanni.ricevuti@unipv.it (G.R.); 3Villa Mafalda Clinics, Via Monte delle Gioie 5, 00199 Rome, Italy; 4Department of Clinical and Experimental Medicine, University of Messina, 98122 Messina, Italy; berrettama@gmail.com; 5Department of Drug Sciences, School of Pharmacy, University of Pavia, 27100 Pavia, Italy; 6Department of Neurosciences, Biomedicine and Movement Sciences, University of Verona, 37129 Verona, Italy

**Keywords:** ME/CFS, ozone, oxygen-ozone therapy, fatigue, clinical trial

## Abstract

(1) Background: Myalgic Encephalomyelitis/Chronic Fatigue Syndrome (ME/CFS) is a chronic syndrome characterized by fatigue as its major and most outstanding symptom. Previous evidence has supported the ability of ozone to relief ME/CFS related fatigue in affected patients (2) Methods: A number of 200 ME/CFS previously diagnosed patients, (mean age 33 ± 13 SD years) were consecutively treated with oxygen-ozone autohemotherapy (O_2_-O_3_-AHT). Fatigue was evaluated via an FSS 7-scoring questionnaire before and following 30 days after treatment. (3) Results: Almost half (43.5%) of the treated patients evolved their FSS scale from the worst (7) to the best (1) score, assessing the highest improvement from being treated with O_2_-O_3_-AHT. Furthermore 77.5% of patients experienced significant ameliorations of fatigue, of 4–6 delta score. No patient showed side effects, yet experienced long lasting fatigue disappearance, by three months follow up (4) Conclusions: Treatment with O_2_-O_3_-AHT greatly improves ME/CFS related fatigue, aside from sex and age distribution.

## 1. Introduction

Myalgic encephalomyelitis/chronic fatigue syndrome (ME/CFS), which has been thoroughly reviewed very recently [1], is a complex terminology used to identify a multi-systemic, serious and long-term illness characterized by fatigue and debilitating muscular-skeletal pain, which usually compels patients to abruptly shutting down many active aspects of their social lives [2,3,4]. Fatigue is a symptom of the utmost importance in ME/CFS and is usually employed to test the general development of ME/CFS symptomatology upon a defined therapy [5,6,7,8,9,10,11]. The epidemiology of ME/CFS has been widely investigated [12,13,14,15]. Previous studies conducted in Italy as forerunner approaches in treating fatigue date back to nineties [5,6,11,16]. In Italy, a recent study performed on 82 CFS patients living in the Northern areas of the country, reported that the mean age was 32 years, quite half of these patients showed early symptoms between 24 and 40 years and ME/CFS was prevalently (3:1) observed in female subjects [15]. Yet, evaluating a correct prevalence of ME/CFS is a burdensome task, as ME/CFS diagnosis is particularly difficult and often patients with fatigue and other clinical signs are misunderstood with other chronic illnesses [17,18]. The recent diagnostic criteria for ME/CFS from the Centers for Disease Control and Prevention, the so-called IOM 2015 Diagnostic Criteria, which updated the CDC-1994 guidelines [19], encompassed symptoms and signs such as a severe and chronic fatigue usually lasting longer than six months, as well as the presence of at least four of the following physical symptoms: unrefreshing sleep, impaired concentration, attention or memory, post-exertional malaise, headaches, muscular-skeletal pain and polyarthralgia, tender lymph nodes and sore throat [20,21,22,23]. ME/CFS diagnosis is therefore quite completely based on the clinical evidence, where long-lasting, debilitating fatigue represents the major and most outstanding symptom [24,25]

The difficulty in reaching a sound and proper diagnostic evaluation of ME/CFS, hampers somehow the possibility to find an effective therapy for ME/CFS [26]. Commendable attempts were addressed in the past, even by our group [5,7,11], yet ME/CFS remains so far a very complex task for clinics. Individuals with ME/CFS should be investigated for concurrent depression, pain, and even sleep disturbances. To date, the only available treatment options include a kind of cognitive behavior therapy, including graded motor exercise rehabilitation, approaches that have been demonstrated to only slightly improve fatigue and ameliorating also social adjustment, anxiety, and post-exertional malaise. However, no pharmacologic medicine therapies have been proven thoroughly effective, to date. 

Encouraging attempts in treating ME/CFS fatigue with oxygen-ozone autohemotherapy (O_2_-O_3_-AHT) were successfully accomplished by our groups and others [27,28,29]. Oxygen-ozone therapy is able to modulate many complex aspects of immunity, the majority of which underlie the pathogenetic mechanisms causing ME/CFS [30,31]. Interestingly, as occurring in COVID-19, where O_2_-O_3_-AHT resulted particularly effective in treating the oxidative stress associated with the sickness, ME/CFS too might have an oxidative stress causative pathogenesis [32,33]. This evidence, associated with the increasing awareness that ozone is able to modulate inflammation by targeting the oxidative stress signaling [32] and by previous encouraging outcomes on ME/CFS [27], suggested us to treat fatigue in patients with O_2_-O_3_-AHT.

In this paper we treated fatigue symptomatology in a large cohort of patients diagnosed as ME/CFS by adopting O_2_-O_3_-AHT. Results are herein described and discussed. 

## 2. Materials and Methods

### 2.1. Patients’ Recruitment

A number of 224 outpatients coming as a whole to the different clinical centers of Pordenone and Gorle (Bergamo) were recruited on the basis of the eligibility criteria assessed for the present research study, 200 entered the study, 19 did not follow up the study design and were excluded from the investigation. 5 formally accepted the research study but never started for different individual reasons. Age distribution reported a mean of 33.08 ± 13.50 SD years [CI_95_ = 31.20–34.97] and a median of 33.14 years, the cohort was composed of 69 males (34.5%), so the majority of patients was represented by female subjects. All patients were thoroughly informed of the therapy protocol and about the use of data for research purposes, according the recommendations of the Helsinki Declaration.

### 2.2. Inclusion and Exclusion Criteria 

Inclusion criteria were represented by outpatients referring to our clinical healthcare previously diagnosed for ME/CFS [23,34], suffering from fatigue, having accepted and signed the informed consent for therapy and moreover enabled and agreed in sharing data for research. Exclusion criteria were represented by patients with chronic active inflammatory pathologies or neurodegenerative and psychiatric disorders, SARS-CoV2 positivity, having experienced COVID-19 within the previous 6 months, included in the suspicion of a post-COVID symptomatology, intaking pharmaceutical drugs within the previous 72 h, having been diagnosed for autoimmune disorders aside from ME/CFS, having cancer, if pregnant women. 

### 2.3. Sample Size

Sample size was calculated to achieve an error range of about 10%. Referring to a population proportion of 51%, forecast data resulted in a 13.86% error with 50 patients, whereas 9.80% (<10%) with 103 patients, therefore 200 patients were within the minimal sample size with *p* < 0.001. The Cohen d test for the two separate groups, i.e., before O_2_-O_3_-AHT and following O_2_-O_3_-AHT was successful, (*p* = 0.004323 or Hedges’ *g* value). Moreover, Glass’ delta *p* = 0.012444 (*p* < 0.02).

### 2.4. Patients’ Evaluation of the Fatigue Symptomatology

An anamnestic interview and complete visitation of about 40–60 min was performed. Fatigue was the major symptom evaluated in the study as able to highlight the patient’s overall clinical status in the most sound and reliable approach, due to its optimal performance features, stability over time and scant possibility to be shadowed by other minor symptoms. Each patient was asked to respond to a 7-scoring system Fatigue Severity Scale (FSS) before undergoing therapy and one month following therapy [35,36]. Results were collected as scores and statistically evaluated.

### 2.5. Patients’ Treatment with Oxygen Ozone Autohemotherapy (O_2_-O_3_-AHT)

Patients underwent not less than two weekly sessions of major oxygen-ozone autohemotherapy, according to the protocol previously assessed by the Italian Society of Oxygen-Ozone Therapy (SIOOT) [27]. For each O_2_-O_3_-AHT session a treatment time of 30 min was held. Improvements are not immediate but can be observed in the post-treatment period, from 1 week to 1 month following at least one oxygen-ozone therapy session. Therefore, one cannot calculate if more sessions are made, much higher is the improvement. Ozone does not act in a cumulative way but elicits subtle and complex mechanisms in the patient’s physiology. A volume of 200 mL of blood was withdrawn from any patient and collected in a CE certified SANO3 bag, then immediately treated with 45 μg/mL of ozone in a O_2_-O_3_ mixture, continuously regulated and tuned by the instrumental device Multioxygen Medical 95 CPS (Gorle, BG, Italy). The Multioxygen^®^ Medical 95 is disposed as an outpatient unit for O_2_-O_3_ therapy, allowing operators to customize the gas mixture according to the clinical request. Actually, the O_2_-O_3_ mixture generator is managed by a microprocessor that ensures the precision of the ozone delivery, once the O_3_-O_2_ mixture amount is selected by the operator. Therefore, it is possible to customize the treatment by selecting the ozone concentration in a continuous range from 1 to 100 μg of O_3_. Then the 200 mL of ozonized blood was reintroduced into the circulatory blood directly, from the sterile bag. Therefore, the therapy required an ozone generator, medical grade oxygen, a sterile syringe and a certified bag endowed with an intravenous cannula, for the O_2_-O_3_-AHT. If following one week from the O_2_-O_3_ -AHT treatment, the FSS delta score was 0, a second session of O_2_-O_3_-AHT will be performed with 150 mL treated blood at 45 μg/mL of ozone in a O_2_-O_3_ mixture. A median of 155 mL, CI_95_ = 135–245 mL of blood was ozonized and reintroduced for each treated patient [27,37]. Patients were followed up at 30 days subsequently to the second O_2_-O_3_-AHT session and asked to complete the FSS questionnaire, as previously agreed.

### 2.6. Statistics 

Data were collected and expressed as mean ± standard deviation, for quantitative values. Sample size was evaluated by assessing data and forecasting evaluations with Cohen d test and a Glass’ delta. Statistical inference, if any, was evaluated following non-parametric tests. Scores were evaluated by a Kruskall-Wallis test with *p* < 0.05. Data were elaborated with a SPSS v 24 software and Stata software for graphs.

## 3. Results

Table 1 summarizes the results obtained in the present study. 

When treated with O_2_-O_3_-AHT fatigue symptoms within the first one-two weeks ameliorated from a score value of 7 (meaning the worst) to 1 (meaning the best, i.e., completely absent symptoms) in almost half of the oxygen-ozone treated patients (43.5%), whose only 24.64% were males and more than 75% females.

The percentage of patients experiencing a delta scoring of 4–6 points were higher than 75%. Anyway, only 5% reported very small, quite negligible improvements in fatigue symptoms. Age distribution did not show any difference in the whole randomized patients’ population (*p* = 0.38765), therefore age did not affect significantly the results of this investigation. About 6.5% of patients diagnosed with ME/CFS, with fatigue symptoms and treated with O_2_-O_3_-AHT were teen agers. The Kruskall-Wallis (KW test) for this cohort of subjects was 18.7778 (*p* = 0.00001). Neither age nor sex distribution affected the correct evaluation of the FSS questionnaire, as KW tests were significant at *p* < 0.05 (Table 1)), and when this ratio was referred to the number of subjects recruited on the total, the rate of males responding to the therapy for about 60% (59.48%) was 1.5 higher than females, responding for about 75% (74.58%) to the therapy. 

This might indicate that male patients are particularly sensitive to O_2_-O_3_-AHT despite their age range and that female are much more responsive but with higher variability. Quite 80% of patients responded optimally to O_2_-O_3_-AHT (77.5%) and more than half (51%) responded with the maximal FSS delta score. Figure 1 shows the impressive difference in fatigue score before and after the O_2_-O_3_-AHT. 

No patient reported adverse manifestation upon O_2_-O_3_-AHT and following the next 3 months from the treatment.

## 4. Discussion

The results here described assess that O_2_-O_3_-AHT is able to relief fatigue in almost half of the whole cohort of ME/CFS patients. The study of ME/CFS was pioneered in Italy by Tirelli and coworkers [11], when several theories about the concurrent viral origin of ME/CFS and the immune impairment in T cell activation and NK cell functional depression were addressed [11]. Immune modulation, even via the tuning of cell survival with the oxidative stress response, might be a solution to address ME/CFS symptomatology, mainly represented by fatigue, and ozone may be a possible approach [5,6,27]. So far, no reliable explanation has been forwarded to elucidate how ozone is able to restore wellness in patients suffering from ME/CFS-related fatigue [27]. The recent evidence associating COVID-19 with ME/CFS pathogenesis as a manifestation of an impaired stress response is particularly intriguing [33]. The ability of ozone to modulate the Nrf2/Keap1/ARE system and the NO/eNOS signaling pathways may provide insightful suggestions to highlight possible mechanisms underlying the effect of O_2_-O_3_-AHT in ME/CFS-caused fatigue symptomatology [27]. During ME/CFS mitochondria functionality may be greatly disturbed, so generating an impairment in many mitochondria-related activities, including ROS signaling [38] and in inflammation disorders [39,40]. ME/CFS is characterized by an increase in CD4^+^CD25^+^Foxp3^+^ T regulatory (Treg) cells [41], which may be modulated and lowered by ozone [42]. Another impairment in ME/CFS regards Th17 cells. The CCR6^+^ Th17 cells in ME/CFS were reported to secrete less IL-17 respect to healthy subjects, circulating cell frequency is significantly lower, whereas ozone can restore their amount, even with IL-17A [43,44]. The immune micro-environment in ME/CFS is of the utmost importance to elucidate how ME/CFS- associated fatigue develops and how treating the same [9,10,11]. To date, the mechanisms with which O_2_-O_3_-AHT may restore physical activity and relief fatigue-related discomfort and myalgia or arthralgia, is far to be fully elucidated. A dysregulated Treg immune response and an impaired Th2 and Th17 cytokine pattern, may be modulated and tuned by ozone and its oxidized metabolites. This activity is closely linked with the cell anti-oxidant endowment and the machinery deputed to the optimal stress response. Noticeably, the major ozone-induced blood byproduct, the 4-hydroxynonenal (4-HNE) induces the thioredoxin reductase 1, via the Nrf2 pathway, then increasing the level of Tregs [45,46]. 

Although the scant evidence able to tailoring a speculative hypothesis on how O_2_-O_3_-AHT can relief fatigue in ME/CFS patients may regard cells and functional models either than ME/CFS, the ability of ozone to modulate immunity and inflammation via the Nrf2 system is particularly documented [32]. 

Furthermore, ozone may regulate nitric oxide (NO) and the activity of the endothelial nitric oxide synthase (eNOS) [32]. Yet, ME/CFS patients seem to have normal NO alongside with normal IL-6 levels, before and after a physical exercise upon fatigue symptoms but exhibit high levels of F2-isoprostanes, which are oxidative stress biomarkers, probably quenched by the activity of ozone on the Nrf2/Keap1/ARE system [47,48,49]. 

The ability of ozone to modulate the complex interplay between oxidative stress and chronic inflammation may be the key to comprehend how ME/CFS patients felt a rapid relief in their fatigue symptoms following O_2_-O_3_-AHT.

## 5. Conclusions

Patients suffering from ME/CFS fatigue and treated with O_2_-O_3_-AHT experienced rapid relief of their symptoms. Further insights are needed to elucidate the mechanism by which about 95% of treated patients recovered their ability to move without pain and discomfort.

## Figures and Tables

**Figure 1 jcm-11-00029-f001:**
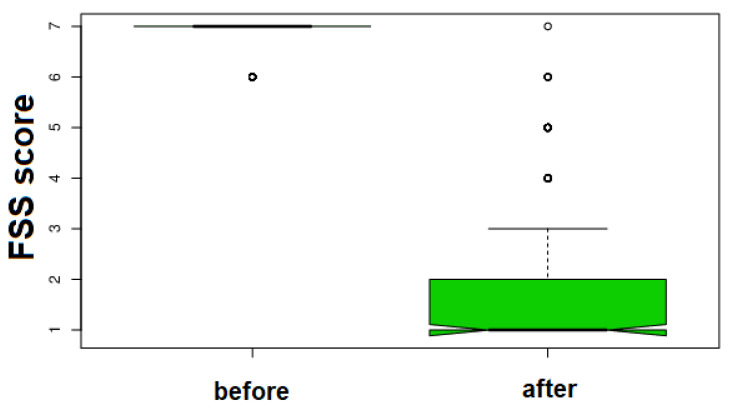
Box plot representation of the impressive change in fatigue symptoms at 30 days after oxygen-ozone autohemotherapy in the 200 patients treated with O_2_-O_3_-AHT.

**Table 1 jcm-11-00029-t001:** Results of fatigue treatment in ME/CFS patients with O_2_-O_3_-AHT (mean ± SD).

Before Treatment		After Treatment				KW Test Delta % Rate				*p*	
All patients											
6.825 ± 0.381		2.085 ± 1.503				H = 293.6672 Δ% = 69.45 0.34				<0.0001	
Male patients											
6.725 ± 0.450		2.725 ± 1.781				H = 54.4956 Δ% = 59.48 0.86				<0.0001	
Female patients											
6.878 ± 0.329		1.748 ± 1.211				H =195.388 Δ% = 74.58 0.57				<0.0001	
Statistical and FFS score data											
Mean age (all): 33.085 ± 13.503 SD				Mean age (males): 37.406 ± 13.958 SD				Mean age (females): 30.809 ± 12.730 SD			
7 to 7 = 1	7 to 6 = 4		7 to 5 = 17		7 to 4 = 9		7 to 3 = 5		7 to 2 = 42		7 to 1 = 87
6 to 6 = 0	6 to 5 = 6		6 to 4 = 3		6 to 3 = 0		6 to 2 = 11		6 to 1 = 15		5 to 1 = 0

Sex distribution: Males = 69; Females = 131 Age distribution: <18 years old = 13; from 18 to 29 years old = 89; ≥30 years old = 98 Spearman Rank Correlation 0 *p* > 0.05 (n.s) *p* = 0.33356, rho = −0.0687, S = 1424930. 3283. KS test for ages *p* = 0.38765 (n.s.). Higher score delta 7 to 1 = 43.5%. Best performance: 77.5%; Excellence: 51%.

## Data Availability

Not applicable.

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
