# Peer review of "Patients with Myalgic Encephalomyelitis/Chronic Fatigue Syndrome (ME/CFS) Greatly Improved Fatigue Symptoms When Treated with Oxygen-Ozone Autohemotherapy"

_jcm, 2021, doi:10.3390/jcm11010029_

Round 1
Reviewer 1 Report
This study builds on prior work from the same group with a larger number of patients and more rigorous study design. The group’s previous report, published in in 2018 demonstrated symptom improvement in a cohort of 65 patients suffering from chronic fatigue syndrome. This new study includes 200 patients and involves a more rigorous statistical analysis of the data. As far as I can tell, the treatment approach between the two studies was the same, however, a key details are missing in the current report, including duration of therapy. This information needs to be included. It would be especially helpful to show the relationship (if any) between the number of times each patient was treated and their reported improvement in fatigue. Also, the authors are encouraged to provide a more detailed clinical protocol in general to enable others to reproduce and expand upon their work. If oxygen/ozone autohemotherapy for a relative short duration can provide significant symptom relief to CFS patients, that would represent a major step forward in the field. However, the practicality of the treatment is currently unclear and the duration of symptom relief has not been addressed. Finally, reference 45 cites work with MS patients that demonstrated decreased IL-17 expression following ozone therapy which does not support the argument being made that ozone can restore IL17 in CFS patients.
Author Response
Point by point rebuttal to the Reviewer’s Comments:
Reviewer 1
This study builds on prior work from the same group with a larger number of patients and more rigorous study design. The group’s previous report, published in in 2018 demonstrated symptom improvement in a cohort of 65 patients suffering from chronic fatigue syndrome. This new study includes 200 patients and involves a more rigorous statistical analysis of the data. As far as I can tell, the treatment approach between the two studies was the same, however, a key details are missing in the current report, including duration of therapy. This information needs to be included.
Authors’ rebuttal: according to the SIOOT protocol each oxygen-ozone therapy session lasts 30 minutes.
It would be especially helpful to show the relationship (if any) between the number of times each patient was treated and their reported improvement in fatigue.
Authors’ rebuttal: Improvements are not immediate but can be observed in the post-treatment period, from 1 week to 1 month following at least one oxygen-ozone therapy session. Therefore, one cannot calculate if more sessions are mede, much higher is the improvement. Ozone does not act in a cumulative way but elicits subtle and complex mechanisms in the patient’s physiology. This sentence was added in the paper
Also, the authors are encouraged to provide a more detailed clinical protocol in general to enable others to reproduce and expand upon their work. If oxygen/ozone autohemotherapy for a relative short duration can provide significant symptom relief to CFS patients, that would represent a major step forward in the field. However, the practicality of the treatment is currently unclear and the duration of symptom relief has not been addressed.
Authors’ rebuttal: This part was added
A volume of 200 ml of blood was withdrawn from any patient and collected in a CE certified SANO3 bag, then immediately treated with 45 μg/ml of ozone in a O2-O3 mixture, continuously regulated and tuned by the instrumental device Multioxygen Medical 95 CPS (Gorle, BG, Italy). The Multioxygen® Medical 95 is disposed as an outpatient unit for O2-O3 therapy, allowing operators to customize the gas mixture according to the clinical request. Actually, the O2-O3 mixture generator is managed by a microprocessor that ensures the precision of the ozone delivery, once the O3-O2 mixture amount is selected by the operator. Therefore, it is possible to customize the treatment by selecting the ozone concentration in a continuous range from 1 to 100 μg of O3. Then the 200 ml of ozonized blood was reintroduced into the circulatory blood directly, from the sterile bag. Therefore, the therapy required an ozone generator, medical grade oxygen, a sterile syringe and a certified bag endowed with an intravenous cannula, for the O2-O3-AHT. If following one week from the O2-O3 -AHT treatment, the FSS delta score was 0, a second session of O2-O3-AHT will be performed with 150 ml treated blood at 45 μg/ml of ozone in a O2-O3 mixture. A median of 155 ml, CI95 = 135–245 ml of blood was ozonized and reintroduced for each treated patient.
Finally, reference 45 cites work with MS patients that demonstrated decreased IL-17 expression following ozone therapy which does not support the argument being made that ozone can restore IL17 in CFS patients.
Authors’ rebuttal: Reference revised
Reviewer 2 Report
This manuscript is devoted to the study of myalgic encephalomyelitis/ chronic fatigue syndrome (ME/CFS), which is a multi-systemic, long-term and debilitating disease, one of the main symptoms of which is prolonged fatigue. As treatment options for chronic fatigue and ME/CFS in general are still limited, the current study is very topical.
The authors already previously had described the possibility of using oxygen-ozone autohemotherapy (O2-O3-AHT) to reduce fatigue in patients with ME/CFS. In this work, the authors have studied the reduction of fatigue in 200 ME/CFS patients by treating them with O2-O3-AHT.
The manuscript is clearly written in good English so I have only a few reservations:
Lines 90-91 – In exclusion criteria authors have mentioned “...immune chronic ailments such as autoimmunity...”, however, the authors should note that ME/CFS is also considered an autoimmune disease:
- Blomberg J, Gottfries C-G, Elfaitouri A, Rizwan M, Rosén A. Infection Elicited Autoimmunity and Myalgic Encephalomyelitis/Chronic Fatigue Syndrome: An Explanatory Model. Front Immunol., 15 February 2018; https://doi.org/10.3389/fimmu.2018.00229
- Sotzny F, Blanco J, Capelli E, Castro-Marrero J, Steiner S, Murovska M, Scheibenbogen C. Myalgic Encephalomyelitis/Chronic Fatigue Syndrome - Evidence for an autoimmune disease. Autoimmun Rev. 2018 Apr 7. https://www.sciencedirect.com/science/article/pii/S1568997218300880
Lines 134, 141, 188, and 196 – “CFS” should be changed to “ME/CFS”.
Lines 138-140 – The sentence is not understandable (at least for me) and needs more explanation.
Lines 150-151 – “dot” should be deleted and the sentence corrected.
Author Response
Reviewer 2
This manuscript is devoted to the study of myalgic encephalomyelitis/ chronic fatigue syndrome (ME/CFS), which is a multi-systemic, long-term and debilitating disease, one of the main symptoms of which is prolonged fatigue. As treatment options for chronic fatigue and ME/CFS in general are still limited, the current study is very topical.
The authors already previously had described the possibility of using oxygen-ozone autohemotherapy (O2-O3-AHT) to reduce fatigue in patients with ME/CFS. In this work, the authors have studied the reduction of fatigue in 200 ME/CFS patients by treating them with O2-O3-AHT.
The manuscript is clearly written in good English so I have only a few reservations:
Lines 90-91 – In exclusion criteria authors have mentioned “...immune chronic ailments such as autoimmunity...”, however, the authors should note that ME/CFS is also considered an autoimmune disease:
- Blomberg J, Gottfries C-G, Elfaitouri A, Rizwan M, Rosén A. Infection Elicited Autoimmunity and Myalgic Encephalomyelitis/Chronic Fatigue Syndrome: An Explanatory Model. Front Immunol., 15 February 2018; https://doi.org/10.3389/fimmu.2018.00229
- Sotzny F, Blanco J, Capelli E, Castro-Marrero J, Steiner S, Murovska M, Scheibenbogen C. Myalgic Encephalomyelitis/Chronic Fatigue Syndrome - Evidence for an autoimmune disease. Autoimmun Rev. 2018 Apr 7. https://www.sciencedirect.com/science/article/pii/S1568997218300880
Authors’ rebuttal. Thank you for the valuable comment. We meant other autoimmune diseases not included in the ME/CFS diagnosis. The sentence was revised accordingly.
Lines 134, 141, 188, and 196 – “CFS” should be changed to “ME/CFS”.
Authors’ rebuttal: Done
Lines 138-140 – The sentence is not understandable (at least for me) and needs more explanation.
Authors’ rebuttal. The sentence was revised as follows:
When treated with O2-O3-AHT fatigue symptoms within the first one-two weeks ameliorated from a score value of 7 (meaning the worst) to 1 (meaning the best, i.e. completely absent symptoms) in almost half of the oxygen-ozone treated patients (43.5%), whose only 24.64% were males and more than 75% females.
Lines 150-151 – “dot” should be deleted and the sentence corrected.
Authors’ rebuttal: Done